# GENERATING MOBILITY TRAJECTORIES WITH REINFORCEMENT LEARNING-ENHANCED GENERATIVE PRE-TRAINED TRANSFORMER

## ABSTRACT

Mobility trajectories are essential for understanding urban dynamics and enhancing urban planning, yet access to such data is frequently hindered by privacy concerns. This research introduces a transformative framework for generating large-scale urban mobility trajectories, employing a novel application of a transformer-based model pre-trained and fine-tuned through a two-phase process. Initially, trajectory generation is conceptualized as an offline reinforcement learning (RL) problem, with a significant reduction in vocabulary space achieved during tokenization. The integration of Inverse Reinforcement Learning (IRL) allows for the capture of trajectory-wise reward signals, leveraging historical data to infer individual mobility preferences. Subsequently, the pre-trained model is fine-tuned using the constructed reward model, effectively addressing the challenges inherent in traditional RL-based autoregressive methods, such as long-term credit assignment and handling of sparse reward environments. Comprehensive evaluations on multiple datasets illustrate that our framework markedly surpasses existing models in terms of reliability and diversity. Our findings not only advance the field of urban mobility modeling but also provide a robust methodology for simulating urban data, with significant implications for traffic management and urban development planning.

## 1 INTRODUCTION

Urban mobility trajectories, including vehicle routes and human movements, are critical in describing crowd dynamics in urban environments. These trajectories not only provide insights into daily travel patterns (Gonzalez et al., 2008; Alessandretti et al., 2020) but also reflect the underlying socio-economic interactions (Barbosa et al., 2021) and urban planning effectiveness Gaglione et al. (2022). Nonetheless, public access to such data is frequently constrained by privacy concerns, underscored by stringent data protection regulations such as the General Data Protection Regulation (GDPR) of the EU. This limitation makes leveraging mobility data in research and practical applications challenging. Therefore, developing methods to generate reliable and diverse urban mobility trajectory data while preserving privacy is crucial for advancing academic research and enabling diverse trajectory-based applications. Generating trajectories that accurately replicate urban dynamics is a complex task. On one hand, a sufficiently large number of generated trajectories is necessary to reveal clear mobility patterns, necessitating high model efficiency. On the other hand, the intricate urban dynamics require capturing the diverse behaviors of travelers, which demands substantial model capacity.

Recent advances in deep generative modeling have shed light on data-driven approaches for simulating urban mobility. Deep generative models, such as Generative Adversarial Networks (GANs) (Goodfellow et al., 2020), Variational Autoencoders (VAEs) (Kingma, 2013), and diffusion models (Ho et al., 2020), have been employed to handle the complicated and diverse mobility by learning mobility patterns directly from data. Adversarial approaches (Choi et al., 2021; Feng et al., 2020) improve expressiveness but are prone to unstable training dynamics. VAE-based techniques (Chen et al., 2021b) approximate trajectory distributions in a latent space but exhibit mode-covering effects due to regularization. Diffusion-based strategies (Zhu et al., 2023; Wei et al., 2024; Zhu et al., 2024) achieve improved sample quality and training stability by iteratively denoising noise into trajectories, yet often incur high computational cost and lack mechanisms for modeling fine-grained decision-making behaviors observed in urban settings.

Notably, Transformer-based autogressive generation offers a scheme to balance between generation quality with training efficiency (Hsu et al., 2024; Qiu et al., 2024; Lin et al., 2025; Haydari et al., 2024). Nevertheless, previous models often fail to account for the distinctive trajectory preferences within urban areas, which extend beyond typical spatial-temporal crowd patterns. For example, some elderly individuals may prefer a smoother, albeit longer, route. To this end, we develop a more efficient and interpretable framework for generating diverse urban mobility trajectories. Specifically, we highlight four key contributions of our proposed framework for urban mobility trajectory generation: **Efficient generation scheme:** We model urban mobility trajectory generation as a partially observable Markov decision process (POMDP) and leverage the Transformer architecture to enable a resource-efficient pre-trained model. **Trajectory-wise reward modeling:** An inverse reinforcement learning-based approach is introduced to calibrate a reward model, effectively capturing and explaining preferred trajectories. **Reward model-based fine-tuning:** For the first time in urban mobility trajectory generation, we enhance the pre-trained model through fine-tuning guided by an explicit reward model. This approach addresses critical challenges in Transformers, including sparse information representation and long-term credit assignment. **Validation through large-scale experiments:** Our framework is rigorously validated on multiple large-scale urban mobility trajectory datasets, showcasing significant improvements in performance and interpretability.

## 2 RELATED WORK

**Urban mobility generation.** While previous rule-based methods (Isaacman et al., 2012; Simini et al., 2021; Zandbergen, 2014) can handle large-scale trajectory generation at a relatively low cost, they struggle to capture the spatial-temporal characteristics of trajectory data. Recent advancements in computational power and deep learning techniques have significantly enhanced the capacity of generative models to learn from large datasets(Long et al., 2023; Song et al., 2024; Wang et al., 2025). Specifically, Choi et al. (2021) and Feng et al. (2020) utilized adversarial learning to learn urban mobility trajectories generation. Additionally, Chen et al. (2021b) introduced a method where each spatial-temporal point in a trajectory is encoded into a unique identifier, employing a VAE-based approach to train the mapping between latent factors and raw trajectory data. Moreover, in the track of diffusion-based models (Zhu et al., 2023; Wei et al., 2024; Zhu et al., 2024), GPS trajectories as a whole are generated from numerous denoising steps.

**Transformer-based trajectory modeling.** To balance generation quality with training efficiency, Transformer-based architectures (Vaswani, 2017) have become the de facto backbone for trajectory modeling. Early work such as Kim et al. (2023) leverages temporal dependencies driven by human decision-making processes to better capture salient events along a trajectory. To alleviate the computational burden, Liang et al. (2022) introduced an auxiliary loss that provides supervision over all output tokens, significantly accelerating Transformer training. More recently, Haydari et al. (2024) employed a gravity-inspired sampling strategy coupled with RL-based fine-tuning to steer large language models toward semantically coherent and realistic trajectories. In parallel, advances in modeling spatial–temporal correlations have further boosted generation fidelity, as exemplified by recent work in (Hsu et al., 2024; Qiu et al., 2024; Lin et al., 2025).

## 3 PROBLEM

In this paper, we focus on large-scale urban trajectory generation tasks. We define a trajectory $\tau$ as a temporally ordered sequence of road segments $l_i$ in a road network, expressed as $\tau = \{l_{t_0}, l_{t_1}, \ldots, l_{t_k}\}$. The generative model $\mathcal{G}$ generates a trajectory $\hat{\tau}$, based on predefined contextual inputs $c$ (e.g., origin and destination), formulated as $\hat{\tau} = \mathcal{G}(c)$. In our proposed framework, we model the trajectory generation as a sequential decision-making problem following Chen et al. (2021a). In this formulation, at each timestep $t$, the agent observes the current state

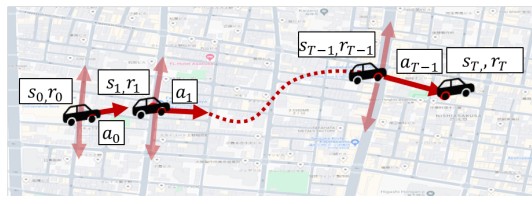

Figure 1: Trajectory generation as a sequential decision-making problem.

$s_t$ (e.g., the current position and contextual factors like traffic conditions), selects an action $a_t$ (e.g.,

determining the next downstream link segment to traverse) and receives a reward $r_t$ representing the quality or efficiency of the chosen action (e.g., whether it arrived at the destination). The trajectory is then generated autoregressively as the agent makes decisions in each segment.

# 4 METHODOLOGY

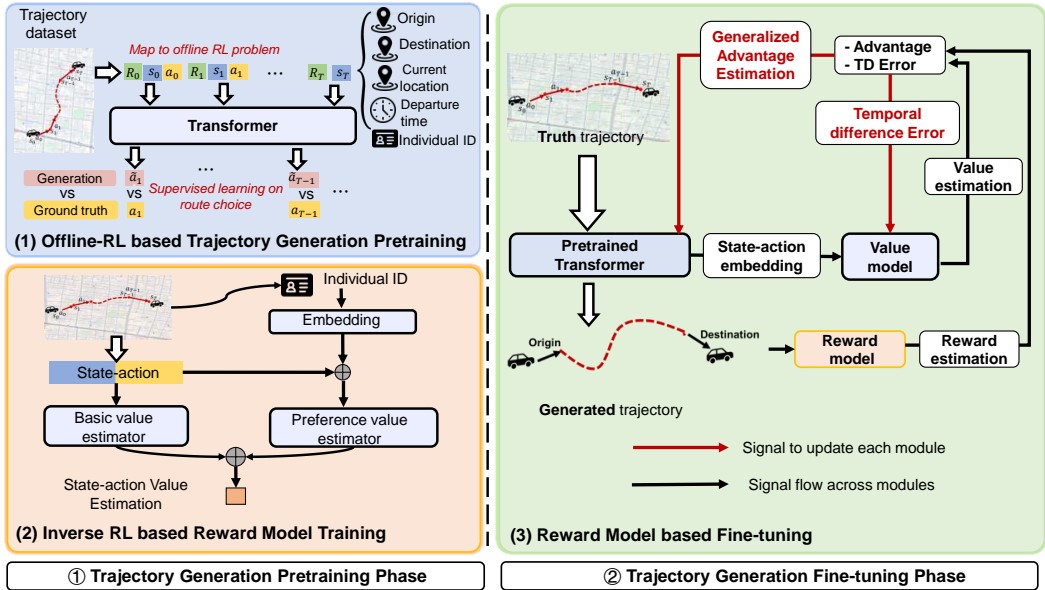

Figure 2: Two-phase framework to enhance pretrained generative model for urban mobility trajectory generation with reinforcement learning (TrajGPT-R). Phase 1: A Generative pre-trained Transformer (GPT) is developed to acquire the general knowledge for generating urban mobility trajectory, meanwhile a reward model is constructed using inverse reinforcement learning to capture trajectory-wise preferences. Phase 2: Reward model-based fine-tuning (RMFT) scheme is introduced to enhance the pre-trained model for better generation reliability and diversity.

## 4.1 OFFLINE-RL BASED TRAJECTORY GENERATION PRETRAINING

In this study, we utilize the Transformer architecture (Vaswani, 2017) as the backbone for trajectory modeling and follow an autoregressive generation scheme. This approach is motivated by two principal factors: Firstly, mobility trajectories share similarities with natural language, such as topological constraints in trajectories that mirror grammatical constraints in text, making the Transformer particularly effective (Zhao et al., 2023). Secondly, the autoregressive generation scheme is suitable for sequential decision-making process modeling, where each decision is predicated on the information available at the current moment.

We employ a token-based approach to represent trajectories in our generation task based on recent advancements in sequential decision modeling with Transformer architectures (Chen et al., 2021a; Wang et al., 2023). Specifically, we utilize three types of tokens—state tokens $s_t$, action tokens $a_t$, and return-to-go tokens $R_t$—to encapsulate the decision-making context at each timestep, as shown in Fig. 2(1). The autoregressive generation mechanism will ensure that each action token is generated in a manner that respects the inherent sequential dependencies of trajectories. In our study, we organize the tokens as: **State token:** Each state token is designed to incorporate spatial and traffic condition information, along with individual ID for personalized context. The spatial context is represented by the current link, origin link, and destination link. Furthermore, we integrate traffic-related features, such as speed and departure intervals into the state token to inform the traffic dynamics. Besides, the individual ID is used to facilitate the learning of nuanced individual preferences. **Action token:** Each action token corresponds to the choice of a downstream link. For example, action token 1 at link $l$ indicates the selection of the first downstream link among all available connections from link

*l.* **Return-to-go token:** This token represents the goal of the current trajectory generation. Unlike previous studies (Chen et al., 2021a), the return-to-go in our tasks is less informative due to the uncertainty in the vehicle's route evaluation, we set the return-to-go as 1 if the vehicle is still en route to its destination and 0 once it arrives. As the trajectory generation in this study follows the autoregressive decision-making process, we incorporate a traditional offline RL scheme (Levine et al., 2020) in the training phase. Specifically, we are training a policy to predict action tokens based on previous tokens. Mathematically, the objective of pretraining is formulated as a cross-entropy loss between the ground truth and the generated decision, defined as:

$$\mathcal{L} = \sum_{t=1}^{T} l\left(a_t, \hat{a}_t\right),$$ (1)

where $\hat{a}_t = \pi_\theta(s_{1:t}, a_{1:t-1}, R_{1:t})$, with $\pi_\theta$ denoting the overall model parameterized by $\theta$. Here, $\hat{a}_t$ represents the action generated at step $t$, given all the previously available tokens, and $T$ denotes the total number of timesteps. The cross-entropy loss $l$ is minimized between the generated action token $\hat{a}_t$ and the ground truth action $a_t$ at each timestep. For the first step, the action token is left blank since there are no prior decisions to consider.

## 4.2 Inverse RL based Reward Model Construction

In previous studies on modeling sequential decision-making using Transformers (Chen et al., 2021a; Wang et al., 2023), the return-to-go token has been introduced as a critical element for highlighting trajectory preferences. However, in the context of trajectory generation, the reward signal is often sparse and poorly defined, which results in a less informative return-to-go token. Furthermore, concerns have been raised regarding the limited capacity for long-term credit assignment in transformer architectures in RL tasks (Ni et al., 2023). Inspired by the benefits of the reward modeling phase in Reinforcement Learning with Human Feedback (RLHF) (Ouyang et al., 2022), this study proposes adapting this scheme to overcome the aforementioned limitations. The rationale is twofold: Firstly, the evaluation from the reward modeling effectively supplements the sparse return-to-go signal in our task. Secondly, by learning to assess the long-term effects of each action, we can leverage offline data to provide a more informative credit-assignment signal.

To this end, we propose constructing a reward model using Inverse Reinforcement Learning (IRL) from the offline data, as depicted in Figure 2(2). The proposed IRL framework is specifically designed to capture both general and individual preferences for routing evaluation through the Basic Value Estimator (BVE) and the Preference Value Estimator (PVE), respectively. Specifically, at each decision step, we partition the available information into two categories: general information (e.g., location, origin-destination, time, and action) and individual-specific messages (e.g., the individual ID). The BVE processes the general information, while the PVE handles the individual-specific messages. The outputs of these two modules are then integrated to produce the state-action value estimates for a specific individual. These estimates are used to evaluate the benefits an individual gains from selecting a particular downstream link in the current state context. In our study, the state-of-the-art IRL approach (Garg et al., 2021) is adapted to learn the state-action value (e.g., Q-function $Q(s, a)$) in our task:

$$\mathcal{J}(Q) = \mathbb{E}_{(s,a)\sim\mathcal{D}_E}\left[\phi\left(Q(s,a) - \gamma\mathbb{E}_{s'\sim\mathcal{P}(\cdot|s,a)}V^*(s')\right)\right] - (1-\gamma)\mathbb{E}_{\rho_0}\left[V^*(s_0)\right],$$ (2)

where $V^*(s) = \log\sum_a \exp\left(Q(s,a)\right)$, $\mathcal{D}_E$ represents the expert demonstration (i.e., historical vehicle trajectories), and $\phi$ is a concave function that serves as a regularizer.

## 4.3 Reward Model-based Fine-tuning

Given the parametric reward model $r_\phi(s_t, a_t)$, we obtain a trajectory-wise reward signal to evaluate the outputs from the pre-trained model. In alignment with the principles of RLHF (Ouyang et al., 2022), we propose enhancing the pre-trained model by fine-tuning it based on the previously constructed reward model $r_\phi(s_t, a_t)$.

As depicted in Figure 2(3), the pre-trained model functions as the parameterized policy $\pi_\theta$, predicting actions based on prior tokens, where an action $a_t$ is sampled according to $a_t \sim \pi_\theta(\cdot|\mathbf{s}_t, \mathbf{R}_t, \mathbf{a}_{t-1})$. Here, $\mathbf{s}_t$, $\mathbf{R}_t$, and $\mathbf{a}_{t-1}$ represent the state, return-to-go, and previous action tokens, respectively,

occurring before timestep $t$. By learning a value model $V_\phi(s_t, a_t)$ based on the reward model, we can evaluate and update the policy $\pi_\theta$ using policy gradient objectives such as Generalized Advantage Estimation (GAE) (Schulman et al., 2015), defined as follows:

$$A_t^{\text{GAE}(\gamma,\lambda)} = \sum_{k=0}^{\infty} (\gamma\lambda)^k \delta_{t+k}, \tag{3}$$

where $V(s_t)$ is the value function estimate at state $s_t$, $\gamma$ is the discount factor, and $\lambda$ is the GAE parameter that balances bias and variance in the advantage estimates. During fine-tuning, the value model is updated with Temporal Difference (TD) errors $\delta_{t+k}$, given by:

$$\delta_t = r_\phi(s_t, a_t) + \gamma V(s_{t+1}) - V(s_t).$$

Specifically, as $A_t^{\text{GAE}(\gamma,\lambda)}$ represents a more stable advantage based on both the long-term value and the reward signal $r_\phi(s_t, a_t)$ over the trajectory, we use it to formulate the final fine-tuning objective:

$$\max_{\pi_\theta} \mathbb{E}_{\mathbf{s},\mathbf{R}\sim D_{\text{pref}}, a\sim\pi_\theta(\cdot|\mathbf{s},\mathbf{R})} \left[ A_t^{\text{GAE}(\gamma,\lambda)} \right] - \beta \mathbb{D}_{\text{KL}} \left[ \pi_\theta(\cdot|\mathbf{s},\mathbf{R}) \parallel \pi_{\text{ref}}(\cdot|\mathbf{s},\mathbf{R}) \right]. \tag{4}$$

Note that we use the penalized term $\beta \mathbb{D}_{\text{KL}} \left[ \pi_\theta(\cdot|\mathbf{s},\mathbf{R}) \parallel \pi_{\text{ref}}(\cdot|\mathbf{s},\mathbf{R}) \right]$ to prevent the updated policy from evolving too abruptly, where $\beta$ is the hyper-parameter weight.

This fine-tuning scheme is expected to improve performance in two primary ways: First, the reward model, trained using IRL, adaptively extracts preference-based information to provide immediate reward signals $r_\phi(s_t, a_t)$ for each generation. Second, by incorporating reward signals with online updating of the value model, the derived fine-tuning objective accounts for the long-term effects at each generation step, addressing the long-term credit assignment gap typically faced by transformer-based pre-trained models in decision-making tasks (Ni et al., 2023).

## 5 EXPERIMENT ANALYSIS

### 5.1 EXPERIMENT SETTING

**Datasets** We consider multiple large-scale urban mobility trajectory datasets to validate the generation performance and applicability, including: *Toyota Dataset:* This dataset consists of 295,488 GPS trajectories collected from Toyota vehicles operating in the Tokyo metropolitan area, covering the period from October 1 to December 31, 2021. *T-Drive Dataset:* (Yuan et al., 2010) This dataset contains GPS trajectories from 10,357 taxis in Beijing, recorded between February 2 and February 8, 2008. *Porto Taxi Dataset:*(Moreira-Matias et al., 2013) This dataset includes GPS trajectories of 441 taxis operating in Porto, Portugal, recorded over one year (from July 1, 2013, to June 30, 2014). For each dataset, we adopt an 80/20 split for training and evaluation.

**Metrics** To comprehensively evaluate the generated results, eight generation metrics have been used to evaluate the performance of the model. These include Jaccard similarity (Jac), Cosine similarity (Cos), BLEU, Jensen-divergence of link distribution (L-JSD), and Jensen-divergence of connection distribution (C-JSD). These metrics are designed to gauge the reliability of the results from a microscopic perspective (i.e. Jac, Cos, and BLEU) and an aggregated perspective (i.e., L-JSD and C-JSD). In addition, the diversity of generations within the urban context is also assessed, particularly through the unigram entropy (UE) and bigram entropy (BE), which measure the diversity of individual links and transitions between links, respectively.

**Model Configurations** All experiments were performed using Python 3.11.8. The deep learning methods are implemented using PyTorch 2.5.0. We run all experiments on a server running Ubuntu 22.04.4, equipped with four NVIDIA RTX A6000 GPUs.

**Baselines** To evaluate the efficacy of our proposed generation framework, we compared it against a diverse set of baseline models that utilize different generation schemes or architectures. The selected baselines include: a statistical method, **Markov** (Korolyuk et al., 1975); a GAN-based method, **TrajGAIL** (Choi et al., 2021); a VAE-based generation method, **TrajVAE** (Chen et al., 2021b); an IRL-based training method, **IQL** (Garg et al., 2021); a diffusion model-based generation approach, **D3PM** (Austin et al., 2021); and two ablation baselines, **TrajGPT** and **TrajGPT-DPO** (Rafailov

et al., 2024). Here, **TrajGPT** refers to the pre-trained phase (i.e., phase 1), while **TrajGPT-DPO** refers to the fine-tuning of the pre-trained model using the fine-tuning scheme without an explicit reward model. The latter two are regarded as ablation baselines for the proposed **TrajGPT-R**, in which RMFT is adopted as the fine-tuning scheme. The details are introduced in the Appendix.

## 5.2 Performance Evaluation

To comprehensively evaluate the performance of our trajectory generation methods, we conducted experiments across three benchmark datasets: Toyota, T-Drive, and Porto. Since these datasets originate from different geographical areas and represent distinct demographics (i.e., the general public for the Toyota dataset and taxi drivers for T-Drive and Porto), this diversity allows us to thoroughly assess the generalization capabilities of our proposed framework.

The experimental results are reported in Table 1. For each dataset, we generate 5,000 trajectories for evaluation. Upon evaluation across all metrics, our proposed framework (e.g., TrajGPT-R) demonstrates superior performance in generating diverse and accurate trajectories. Specifically, it achieves the highest reliability scores, including a Jaccard and cosine similarity, and a BLEU score. It also records the lowest values in L-JSD and C-JSD, indicating minimal distribution divergence. Additionally, the framework attains high entropy scores. Similar trends can be observed in the T-Drive dataset, with TrajGPT-R slightly outperforming other models. It achieved the highest Jac score and competitive results in Cos and BLEU. Both TrajGPT-R and TrajGPT-DPO demonstrate excellent efficiency, tying for the lowest L-JSD and C-JSD scores, which validates the necessity of fine-tuning. The performance across the Porto dataset is consistent with the other datasets, with TrajGPT-R consistently outperforming other models in most metrics. This is particularly notable given the complex urban dynamics represented in the Porto data. The experimental results validate the effectiveness of the proposed TrajGPT-R framework. The enhancements achieved during the fine-tuning phase, supported by the reward model, significantly contribute to its performance, surpassing both traditional and advanced trajectory modeling techniques. Notably, as the three datasets exhibit different trip dynamics (e.g., general public preferences in the Toyota dataset versus taxi driver behaviors in the T-Drive and Porto datasets), the TrajGPT-R framework demonstrates that through fine-tuning with an explicit reward model, we can effectively handle urban mobility trajectory generation across various scenarios.

For a more intuitive examination of the generation results, we visualize the trajectories generated by our framework and compare them with those from ablation baselines. Fig 3 displays 5,000 generated and ground-truth trajectories based on the Toyota Dataset within the core area of Tokyo, Japan. We observe that the fine-tuning phase through DPO or RMFT significantly enhances the TrajGPT capability to accurately generate trajectories in sparsely populated areas (e.g., the north area of the map). This improvement underscores the positive impact of the RL fine-tuning scheme on model generalization, aligning with the findings reported in Tajwar et al. (2024).

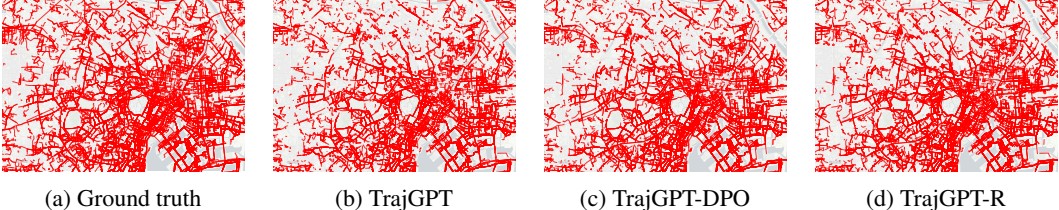

| (a) Ground truth | (b) TrajGPT | (c) TrajGPT-DPO | (d) TrajGPT-R |

Figure 3: Generated and real trajectories based on the Toyota Dataset in the core area of Tokyo.

The robustness of the generation can be further examined along the temporal dimension, as shown in Fig. 4. This analysis reveals how effectively the model captures variations over time and adapts to varying traffic conditions. Specifically, TrajGPT-R demonstrates a consistent ability to generate trajectories that align closely with real-world temporal patterns. These results underscore the model's capacity for handling real-time variabilities and maintaining reliability across extended periods, which is promising for practical data generation for urban planning or traffic management.

Table 1: Method Comparison Across Metrics for Toyota, T-Drive, and Porto Datasets

| Method | Reliability | | | | | Diversity | |
|---|---|---|---|---|---|---|---|
| | Jac(↑) | Cos(↑) | BLEU(↑) | L-JSD(↓) | C-JSD(↓) | UE(↑) | BE(↑) |
| **Toyota Dataset** | | | | | | | |
| **Markov** | 0.198 | 0.291 | 0.008 | 0.340 | 0.692 | 13.40 | 13.75 |
| **TrajVAE** | 0.181 | 0.271 | 0.018 | 0.056 | 0.174 | 12.31 | 12.49 |
| **TrajGAIL** | 0.124 | 0.206 | 0.001 | 0.072 | 0.234 | 12.22 | 12.61 |
| **D3PM** | 0.173 | 0.225 | 0.021 | 0.692 | 0.691 | 12.02 | 12.24 |
| **IQL** | 0.236 | 0.271 | 0.072 | 0.026 | 0.075 | 14.17 | 14.28 |
| **TrajGPT** | 0.390 | 0.455 | 0.225 | 0.028 | 0.070 | 14.07 | 14.41 |
| **TrajGPT-DPO** | 0.499 | 0.556 | 0.341 | 0.021 | 0.052 | 14.44 | 14.75 |
| **TrajGPT-R** | **0.524** | **0.575** | **0.383** | **0.016** | **0.042** | **14.85** | **14.82** |
| **T-Drive Dataset** | | | | | | | |
| **Markov** | 0.225 | 0.428 | 0.000 | 0.007 | 0.137 | 3.66 | 3.63 |
| **TrajVAE** | 0.200 | 0.405 | 0.054 | 0.054 | 0.144 | 8.27 | 9.00 |
| **TrajGAIL** | 0.172 | 0.274 | 0.000 | 0.166 | 0.328 | 8.06 | 9.06 |
| **D3PM** | 0.231 | 0.246 | 0.127 | 0.691 | 0.691 | 7.89 | 8.65 |
| **IQL** | 0.314 | 0.374 | 0.195 | 0.005 | 0.010 | 8.40 | 9.81 |
| **TrajGPT** | 0.617 | 0.567 | 0.317 | 0.005 | 0.014 | 8.06 | 9.87 |
| **TrajGPT-DPO** | 0.612 | **0.573** | 0.333 | 0.005 | **0.011** | 8.49 | 10.12 |
| **TrajGPT-R** | **0.635** | 0.570 | **0.345** | **0.005** | 0.013 | **8.57** | **10.22** |
| **Porto Dataset** | | | | | | | |
| **Markov** | 0.080 | 0.332 | 0.099 | 0.077 | 0.209 | 6.71 | 6.71 |
| **TrajVAE** | 0.080 | 0.320 | 0.158 | 0.093 | 0.226 | 6.63 | 6.71 |
| **TrajGAIL** | 0.102 | 0.247 | 0.050 | 0.021 | 0.035 | 8.46 | 10.47 |
| **D3PM** | 0.083 | 0.333 | 0.259 | 0.111 | 0.225 | 6.81 | 6.82 |
| **IQL** | 0.214 | 0.259 | 0.215 | 0.023 | 0.042 | 7.77 | 8.52 |
| **TrajGPT** | 0.319 | 0.350 | 0.322 | 0.023 | 0.042 | 8.13 | 9.17 |
| **TrajGPT-DPO** | 0.337 | 0.353 | 0.332 | 0.024 | 0.040 | 8.06 | 9.06 |
| **TrajGPT-R** | **0.522** | **0.470** | **0.432** | **0.013** | **0.032** | **10.13** | **10.75** |

## 5.3 INTERPRETABILITY ANALYSIS

In this section, we discuss the mechanism of the proposed framework for mastering trajectory generation through an intuitive and interpretative approach. Specifically, by integrating explicit individual modeling via individual ID embeddings—which potentially encode personal preferences—and employing an autoregressive decision-making scheme, we address the two key questions: **Q1:** How do individual embeddings evolve in different training stages? **Q2:** How does the autoregressive decision-making scheme utilize each token?

Without loss of generality, our analysis and demonstrations is based on the Toyota dataset. 5000 trajectories are generated through a sequential four-phase process using our foundation model. These phases include Initialization, the pre-trained model (TrajGPT), and two fine-tuned models, TrajGPT-DPO and TrajGPT-R, employing different tuning schemes: DPO and RMFT, respectively.

To address **Q1**, we visualize the individual embeddings in a two-dimensional space by applying t-SNE (Van der Maaten & Hinton, 2008) for dimensionality reduction. Prior studies have investigated travel behavior by examining trip entropy (Goulet-Langlois et al., 2017; Huang et al., 2019). To evaluate whether individual embeddings capture routing preferences, we assign a label to each individual based on their route-choice entropy (RCE). Specifically, we model route choice as a tuple consisting of the upstream link, downstream link, and departure time (e.g., a time period within a day). The RCE is computed as $H = -\sum_{i=1}^{n} p_i \log(p_i)$, where $n$ is the number of different tuples and $p_i$ is the probability that tuple $i$ is chosen. The logarithm base used can vary depending on context (e.g., base 2 for binary entropy, or natural logarithm for information measured in nats). Intuitively, a higher RCE indicates a more diverse routing preference, whereas a lower RCE suggests that the individual follows a more consistent routing pattern.

As illustrated in Fig. 5, the individual embeddings in a two-dimensional space exhibit interesting patterns at various training stages. Initially, the embeddings from the initialized model appear disorganized. After Phase 1, the embeddings from the pre-trained model distinctly form two clusters corresponding to lower and higher entropy. This separation underscores the ability of individual

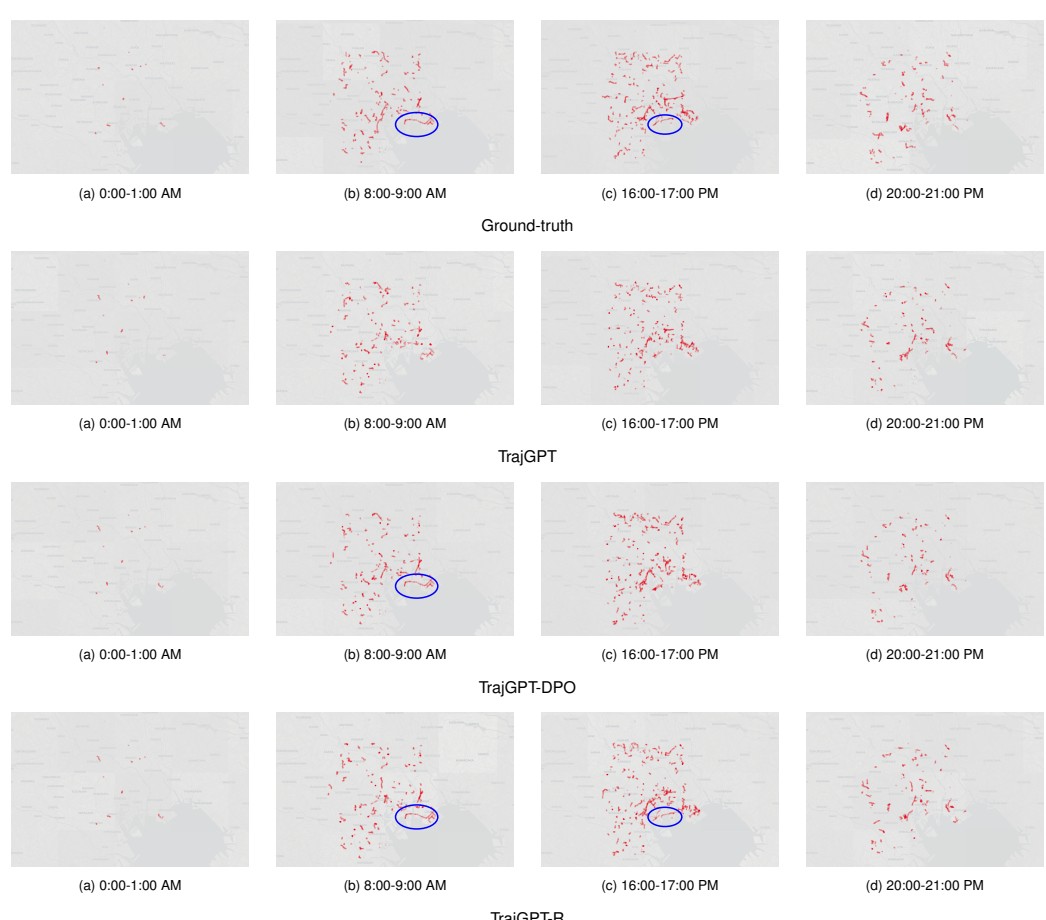

Figure 4: Temporal variation in trajectory generation. We compare generated trajectories across four representative time periods—morning non-peak, morning peak, afternoon peak, and afternoon non-peak—to illustrate temporal dynamics in urban mobility. Fine-tuning strategies can enhance the pretrained model's ability to capture subtle trajectory patterns (i.e., in green circles).

embeddings to capture essential information about travel regularity. More interestingly, while the

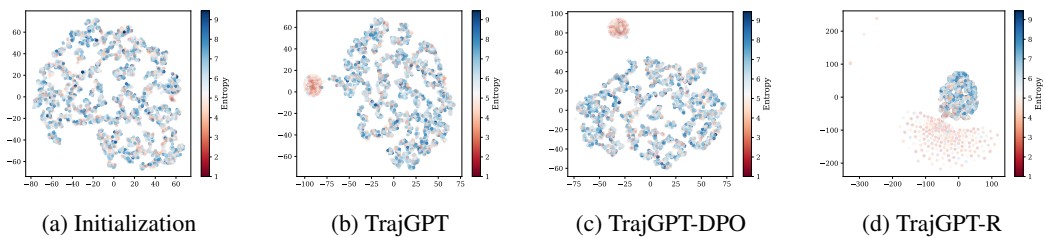

(a) Initialization      (b) TrajGPT      (c) TrajGPT-DPO      (d) TrajGPT-R

Figure 5: Visualization of embeddings projected into 2D space using t-SNE, colored by RCE.

clustering pattern remains largely unchanged after fine-tuning with the DPO scheme, the proposed scheme shows a significant evolution in its clustering (see Fig. 5d). Compared to other stages, the low-entropy cluster from the proposed RMFT becomes more dispersed, suggesting an enhanced capability of the model to differentiate individuals with regular travel patterns. Conversely, the high-entropy cluster is more condensed, indicating that the model effectively isolates irregular travel behavior, treating it akin to noise. These observations suggest that the RMFT can refine the model's ability to discern and categorize individual behaviors based on their travel regularity.

In the next step, we analyze how the model leverages input information for the generation tasks (i.e., **Q2**). To explore these aspects, we first collect and rank attention scores for various tokens at different relative time steps in the generation process. Here, the notations $S@t$, $R@t$, and $A@t$ refer to the state, return-to-go, and action tokens, respectively. Each token occurs at the $t$th relative time step in generating the future action, where a smaller $t$ value indicates a closer location to the current generation output. Take the generation of the $T$th action as an example, $S@0$ refers to the state token immediately before the generation of the $T$th action, and $S@1$ represents the state token one step prior, used in generating the $T$th action.

Note that we limit the analysis to the top 6 ranked attention scores for clarity in the presentation. As shown in Fig. 6, it is observed that, compared to the initialization, the models after different training phases exhibit a more distinct pattern in assigning attention scores. Specifically, the state token immediately preceding each generation step (i.e., $S@0$) consistently receives the highest attention score most of the time. This makes sense as the most immediate observation often provides the most relevant information for decision-making. Besides, this observation aligns with the formulation of trajectory generation as a partially observable Markov decision process, where future actions are primarily determined by the immediate state observation. Furthermore, the proposed scheme leads to a more diverse distribution of attention scores compared to the other two schemes. This observation aligns with previous findings, indicating that the model fine-tuned with an explicit reward model tends to exploit nuanced information.

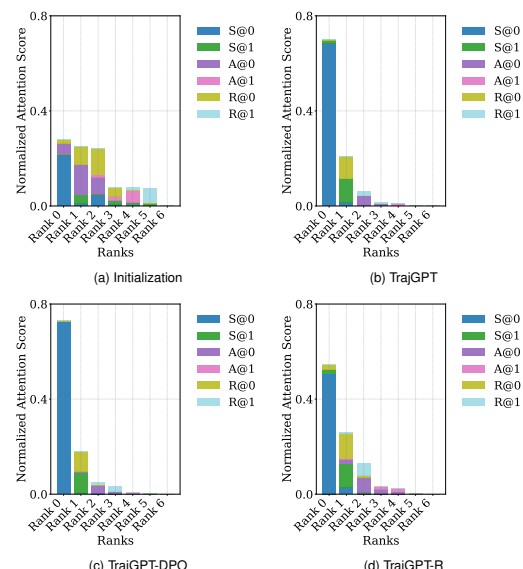

Figure 6: Normalized attention scores for different tokens at various relative positions.

# 6 CONCLUSION

In this study, we introduce *TrajGPT-R* for urban mobility trajectory generation. In the first phase, we model trajectory generation as an offline RL problem using a transformer-based generative model, which benefits from reduced vocabulary complexity. Meanwhile, we construct a trajectory-wise reward model using inverse reinforcement learning (IRL) to capture individual preferences. In the second phase, TrajGPT is fine-tuned with the learned reward model. Extensive experiments on multiple large-scale trajectory datasets show that our approach consistently outperforms baselines in both reliability and diversity. Additionally, interpretability analysis reveals interesting mechanisms of the model, highlighting its robustness and bridging theoretical insights with practical applications.

There are several promising directions for future research. First, compounding errors remain a major challenge in autoregressive generation. Future work could explore advanced sampling methods, training procedures that better align with inference conditions, or architectural improvements to enhance robustness to early errors. Our fully data-driven reward model, unlike traditional RLHF approaches that incorporate human feedback, may introduce bias due to the lack of explicit human guidance. A promising extension is to iteratively adapt the reward model to individual user preferences.

## LLM USAGE STATEMENT

In accordance with the ICLR 2026 policy on LLM usage, we disclose that LLMs (specifically OpenAI's ChatGPT) were employed as a general-purpose writing assistant. The usage was limited to improving grammar, clarity, and LaTeX formatting of the manuscript. All research ideas, experiments, analyses, and conclusions are solely the work of the authors.

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

# A APPENDIX

## A.1 METRIC

**1. Jaccard Similarity (Jac):**

$$\text{Jac}(A, B) = \frac{|A \cap B|}{|A \cup B|},$$

where $A$ and $B$ are sets of elements (e.g., links of each trajectory).

**2. Cosine Similarity (Cos):**

$$\text{Cos}(\mathbf{a}, \mathbf{b}) = \frac{\mathbf{a} \cdot \mathbf{b}}{\|\mathbf{a}\|\|\mathbf{b}\|},$$

where $\mathbf{a}$ and $\mathbf{b}$ are vector representations of trajectories.

**3. BLEU (Bilingual Evaluation Understudy):** (Papineni et al., 2002). BLEU (Bilingual Evaluation Understudy) is traditionally used to evaluate the quality of text generated by machine translation systems. We adapt it to measure trajectory diversity due to the similarity of the trajectory and text. BLEU score calculations involve modified n-gram precisions and a brevity penalty, formulated as:

$$\text{BLEU} = \text{BP} \cdot \exp\left(\sum_{k=1}^{K} w_k \log p_k\right),$$

where $p_k$ is the precision for k-grams, computed as the ratio of the number of matching k-grams in $t_i$ to the total number of k-grams in $t_i$. $w_k$ is the weight for each k-gram precision, typically uniform across different k values. The brevity penalty (BP) is calculated as:

$$\text{BP} = \begin{cases} 1 & \text{if } c > r \\ \exp(1 - r/c) & \text{if } c \leq r \end{cases}$$

where $c$ is the length of the candidate sequence and $r$ is the effective reference corpus length.

**4. Jessen-divergence of Link Distribution (L-JSD):**

$$L\text{-JSD}(P \parallel Q) = \frac{1}{2}\left[D(P \parallel M) + D(Q \parallel M)\right],$$

where $P$ and $Q$ are the probabilities of link segments in the generated data and the ground-truth data. $M = \frac{1}{2}(P + Q)$ and $D$ is the Kullback-Leibler divergence. This metric is used to measure the link distribution proximity between the generated and the real data.

**5. Jessen-divergence of Connection Distribution (C-JSD):**

$$C\text{-JSD}(P \parallel Q) = \frac{1}{2}\left[D(P \parallel M) + D(Q \parallel M)\right].$$

This metric is similar to L-JSD but applied to connection distributions.

**6. Unigram Entropy (UE):**

$$UE = -\sum_{i \in L} p_i \log_2(p_i),$$

where $L$ represents the set of all unique links in the dataset, and $p_i$ is the probability of the $i$-th link ID occurring in the generated trajectories. This metric quantifies the diversity of individual links, reflecting their variety at the most granular level.

**7. Bigram Entropy (BE):**

$$BE = -\sum_{(i,j) \in C} p_{ij} \log_2(p_{ij}),$$

where $C$ denotes the set of all unique consecutive link pairs (connections) in the dataset, and $p_{ij}$ is the probability of the pair $(i, j)$ occurring in the generated trajectories. This metric evaluates the diversity of transitions between consecutive links, providing insight into the local structural variety.

## A.2 MODEL CONFIGURATIONS

This appendix provides the configuration details of the deep learning models implemented using PyTorch for the experiments discussed in the main document. The models are designed to cater to different environments, namely Toyota, T-Drive and Porto datasets. GPT2Model, a variant of the GPT-2 architecture, is served as backbone with a specific configuration for different data.

### ENVIRONMENT-SPECIFIC CONFIGURATIONS

**1. Toyota dataset configuration**

- **Embeddings:** Various embeddings are employed to encode different types of inputs:
  - **Link, Origin, Destination:** Embedded using separate embeddings with a vocabulary size of 262144.
  - **Action, Departure Time:** Action dimensions and departure times are embedded with their respective sizes.
  - **Speed:** Embedded using an embedding layer designed for a range of 120 different speeds.
- **Layer Normalization:** Applied post-embedding to stabilize the learning process.
- **Predictive Outputs:** Includes prediction of actions and optionally states and returns, facilitated by linear transformations and activation functions.

**2. T-Drive Dataset Configuration** Similar to the Toyota configuration with adjustments to embedding sizes for Link, Origin, and Destination, each reduced to a vocabulary size of 16384.

**3. Porto Dataset Configuration** Adapted embedding sizes for Link, Origin, and Destination, reflecting the smaller geographic scope and dataset size with a vocabulary size of 5524.

### COMMON FEATURES ACROSS ENVIRONMENTS

- **Individual Embeddings:** We use word embedding with the actual number of individuals as the vocabulary size.
- **Timestep Embedding:** We use word embedding with a maximum trajectory length as the vocabulary size.

### MODEL TRAINING CONFIGURATIONS

Table 2: Summary of Training Configurations

| Parameter | Toyota | T-Drive | Porto |
|---|---|---|---|
| Weight Decay | 0.05 | 0.02 | 0.05 |
| Embedding Dimension | 512 | 256 | 256 |
| Learning Rate | 0.0005 | 0.0005 | 0.0005 |
| Sub-sample Length | 64 | 12 | 64 |
| Batch Size | 64 | 64 | 128 |
| Attention Layers | 2 | 2 | 3 |

### REPRODUCIBILITY

All experiments are implemented in PyTorch and executed on four NVIDIA H6000 GPUs (40GB each). In accordance with the data policy of Toyota Inc., access to the complete ground-truth trajectory dataset requires a formal application process and cannot be freely released. Nevertheless, two open-source datasets (i.e., T-Drive and Porto), along with our implementation and generated results, will be made publicly available upon acceptance of this work.

ETHICS STATEMENT

This work complies with the ICLR Code of Ethics. The study leverages large-scale, anonymized mobility trajectory datasets for training and evaluation. All data were handled in accordance with strict privacy and data-use regulations. No personally identifiable information (PII) or user-level demographics (e.g., age, gender, home–work identifiers) were used, and no attempt was made to reconstruct or deanonymize individual behavior.

The proposed models are designed for methodological advancement in trajectory generation and urban mobility analysis. While the research may inform applications such as transportation planning or disaster response, it does not involve human subjects or interventions, nor does it seek to predict or profile individuals. Potential risks of misuse, including surveillance or discriminatory practices, are acknowledged; to mitigate these, we restrict our work to anonymized data and emphasize aggregate-level evaluation.

We believe this research contributes positively to society by providing tools that can support urban planning, traffic management, and resilience studies. All ethical, legal, and research integrity standards have been respected throughout this work.

