# OpenReview forum: "Generating Mobility Trajectories with Reinforcement Learning-Enhanced Generative Pre-trained Transformer"
_ICLR.cc/2026/Conference — ICLR 2026 Conference Withdrawn Submission_

### Official Review · Reviewer_M6iS · 2025-10-25

**Soundness:** 2
**Presentation:** 2
**Contribution:** 2
**Rating:** 2
**Confidence:** 4

**Summary:**

This paper proposes a novel framework that models urban mobility trajectory generation as a Partially Observable Markov Decision Process. It first pretrains a Transformer-based trajectory generator, then introduces an IRL module to infer a reward function capturing individual user preferences. Finally, the model is fine-tuned through a reward-guided policy optimization stage. Experiments on three real-world city datasets demonstrate that TrajGPT-R outperforms baselines in both reliability and diversity, while also providing interpretable insights into personalized travel behaviors.

**Strengths:**

1. The paper models Urban Mobility trajectory generation as a POMDP and pretrains a Transformer-based generator.  By introducing an IRL-derived reward model that captures and explains user preferences, the framework logically integrates imitation and reinforcement learning.

2.   The experiments cover three major city-scale datasets and multiple evaluation metrics, demonstrating the method’s robustness across diverse mobility scenarios.

**Weaknesses:**

1.   The paper doesn’t fully discuss the implications or theoretical benefits of treating trajectory generation as a decision-making problem.  The action, defined as the “downstream link ID,” causes the action space to vary with the state—violating standard MDP assumptions.  Conceptually, the framework resembles *graph-sequence generation* more than classical RL.

2.   In Phase-1, the RTG is binary ({1, 0}), merely indicating whether the destination is reached.
   This undermines the core “reward-conditioning” role of the Decision Transformer structure.

3.   The IRL-derived reward function has no clear physical or behavioral meaning, and it remains unverified whether high-reward trajectories truly align with human mobility preferences.

4. The framework (pretraining → IRL → fine-tuning) is more complicated than direct generative baselines.  Merging or simplifying certain stages could make the approach more efficient and deployable.

5.    The paper does not evaluate how the generated trajectories perform in practical downstream tasks, leaving external validity untested.

**Questions:**

What guarantees that the IRL-derived reward function provides meaningful guidance during fine-tuning?
The reward model is learned purely from expert trajectories without explicit behavioral or semantic supervision.  How to ensure that this inferred reward truly represents user preferences rather than overfitting to frequent routes?

---

### Official Review · Reviewer_b4gr · 2025-10-29

**Soundness:** 2
**Presentation:** 3
**Contribution:** 2
**Rating:** 2
**Confidence:** 5

**Summary:**

This paper proposes TrajGPT-R, a generative framework for modeling human mobility trajectories that combines Decision Transformer (DT)-style sequence modeling with Inverse Reinforcement Learning (IRL) and a Reward-Model Fine-Tuning (RMFT) stage inspired by RLHF. The method first pretrains a GPT-like autoregressive policy on discretized trajectory tokens (state, action, and return-to-go), then learns an IRL-based reward model from expert data, and finally fine-tunes the policy via a KL-regularized PPO objective using the learned rewards. Experiments on three large-scale real-world datasets (Toyota, T-Drive, and Porto) show improvements over GAN-, VAE-, diffusion-, and IQL-based baselines on distributional metrics such as Jaccard, Cosine, BLEU, and JSD, along with qualitative interpretability analyses through attention and embedding visualizations. Overall, the work demonstrates the feasibility of adapting LLM paradigms to mobility trajectory generation, though its technical novelty is limited and primarily reflects a combination of existing ideas rather than a fundamentally new learning idea.

**Strengths:**

1. The paper presents a coherent two-phase pipeline: DT pretraining followed by reward-model fine-tuning, that is grounded in recent LLM-based imitation learning methods.

2. Experiments on three large-scale, real-world datasets (Toyota, T-Drive, and Porto) cover diverse mobility contexts and consistently report quantitative gains over multiple baselines, including GAN-, VAE-, diffusion-, and IQL-based approaches.

3. Rich interpretability analysis: The paper includes trajectory visualizations, attention heatmaps, and driver-ID embedding clustering, which together offer qualitative insights into model behavior and learned representations.

**Weaknesses:**

While the paper presents a well-engineered combination of trending paradigms including a GPT-style sequence modeling, IRL reward learning, and RLHF-inspired fine-tuning, the contribution is largely incremental. There is no clear demonstration why it is necessary to combine all the three elements together for trajectory generation.

1. The pretrained DT model relies on a binary reward function and is later fine-tuned using the IRL-derived reward. There is no clear support on why changing the reward function would be necessary, and it is likely to introduce conflicting gradients given the different reward designs. It is also not clear why it is necessary to have an IRL-derived reward fine-tuning the policy net after DT training. A more natural thought would be do the IRL first and use the IRL-derived reward to guide DT training.

2. I also have doubt on the tokenization design when discretizing the continuous, time-varying traffic variables (e.g., speed). The quantization errors and temporal drift could undermine the validity of the state representation. The subsequent vocabulary size is also computationally heavy, and likely leading to data sparsity.

3. There are also missing baselines and ablation studies. A lot of related baselines [1,2,3] are missing to be compared with. In addition, it is also important to study what will happen if training the DT with IRL-derived reward.

[1] Zhu, Yuanshao, et al. "Difftraj: Generating gps trajectory with diffusion probabilistic model." Advances in Neural Information Processing Systems 36 (2023): 65168-65188.
[2] Jiang, Wenjun, et al. "Continuous trajectory generation based on two-stage GAN." Proceedings of the AAAI conference on artificial intelligence. Vol. 37. No. 4. 2023.
[3] Haydari, Ammar, et al. "Mobilitygpt: Enhanced human mobility modeling with a gpt model." arXiv preprint arXiv:2402.03264 (2024).

**Questions:**

1. Why not directly use a learned IRL reward for DT return-to-go signal?

2. How to ensure consistency of the DT objective and the IRL-reward objective for fine-tuning?

3. How to validate that the learned reward aligns with interpretable driving objectives?

---

### Official Review · Reviewer_2AQP · 2025-10-31

**Soundness:** 2
**Presentation:** 2
**Contribution:** 2
**Rating:** 2
**Confidence:** 5

**Summary:**

This paper proposes a two-stage framework called TrajGPT-R, aiming to enhance pre-trained generative models for urban travel trajectory generation using reinforcement learning. In the first stage, a generative pre-trained Transformer (GPT) model is developed to acquire general knowledge for generating urban travel trajectories. Simultaneously, a reward model is constructed using inverse reinforcement learning to capture users' stochastic preferences. In the second stage, a reward model-based fine-tuning (RMFT) scheme is proposed to enhance the pre-trained model, aiming to improve the reliability and diversity of generated trajectories. The proposed method achieves excellent results on multiple datasets.

**Strengths:**

1. Comprehensive experiments were conducted. Experimental results demonstrate that the trajectory quality generated by the proposed method outperforms all selected baseline methods.

2. The authors demonstrate the effectiveness of the method design by comparing the proposed TrajGPT-R with its ablation baselines (TrajGPT and TrajGPT-DPO) through various methods, including trajectory and embedding visualization.

**Weaknesses:**

1. The baselines selected in the paper were all published in 2021 or earlier. This weakens the persuasiveness of the superiority of the proposed method to some extent. Including more recent and relevant trajectory generation baselines, such as DiffTraj[1], would more strongly validate the superior performance of the proposed method.
[1] Zhu, Yuanshao, et al. "Difftraj: Generating gps trajectory with diffusion probabilistic model." Advances in Neural Information Processing Systems 36 (2023): 65168-65188.

2. On the T-Drive dataset, the proposed TrajGPT-R does not achieve the optimal JSD metric. Furthermore, compared to the original TrajGPT ablation variant, TrajGPT-R shows negligible improvement in the JSD metric, which contradicts the results of the other two datasets. Could the authors conduct further analysis on this? Clarifying whether this is a limitation of the method itself or a characteristic of the T-Drive dataset would help strengthen the argument for the effectiveness and robustness of the method.

3. In Section 5.3, “Interpretability Analysis,” the paper demonstrates through visualization that the low-entropy clusters generated by the proposed RMFT method are more dispersed, indicating that it can better distinguish individuals with regular travel patterns. Could the authors provide more direct experimental evidence to support this claim? For example, performance metrics for trajectories generated specifically for user groups with regular travel patterns would provide a more direct and convincing argument.


Other issues:

1. The abstract and conclusion of the paper mention a reduction in "lexical complexity." I don't fully understand this. Could the authors further explain how this is reflected in the proposed methodology?

2. There appears to be a small spelling error in Figure 4; it should refer to the "blue circle."

3. For trajectory visualization, it is suggested to include comparisons with the strongest baseline, rather than simply showing visualizations of the TrajGPT series. This would make the qualitative results more compelling.

**Questions:**

See weaknesses

---

### Official Review · Reviewer_adT2 · 2025-10-31

**Soundness:** 2
**Presentation:** 2
**Contribution:** 2
**Rating:** 2
**Confidence:** 3

**Summary:**

This paper presents a two-phase framework called TrajGPT-R for generating urban mobility trajectories. The authors formulate the trajectory generation problem as an offline reinforcement learning task and adopt a Transformer architecture for autoregressive generation. In the first phase, the model acquires generative capability through offline RL pretraining; subsequently, a reward model is constructed based on inverse reinforcement learning (IRL) to capture individual mobility preferences. In the second phase, the model is fine-tuned using the learned reward model, which is claimed to improve the diversity and reliability of generated trajectories. The paper conducts experiments on three trajectory datasets—Toyota, T-Drive, and Porto—and compares the proposed method against VAE-, GAN-, diffusion-, and RL-based baselines using multiple similarity and diversity metrics (e.g., Jaccard, BLEU, JSD, and entropy). The results indicate that TrajGPT-R outperforms most baselines across these metrics. In addition, the authors provide interpretability analyses that visualize how individual embeddings and attention patterns evolve during different training stages. Overall, the paper aims to generate trajectory data consistent with urban mobility characteristics by integrating Transformer-based modeling with reinforcement learning mechanisms, addressing the challenge of mobility data generation under privacy constraints.

**Strengths:**

1. The paper proposes a two-phase framework (TrajGPT-R) that combines offline reinforcement learning with inverse reinforcement learning–based reward modeling, representing an interesting attempt to bridge sequential decision-making and generative modeling for urban mobility trajectory generation.

2. The authors evaluate the proposed approach on multiple large-scale real-world datasets (Toyota, T-Drive, and Porto) and compare it with a diverse set of baselines (VAE-, GAN-, diffusion-, and RL-based models), providing quantitative results across multiple metrics as well as interpretability analyses through embedding visualization and attention inspection.

**Weaknesses:**

1. Despite claiming a new two-phase framework, the paper mainly combines existing ideas—offline RL, IRL, and Transformer-based trajectory generation—without introducing substantial theoretical or algorithmic innovation. The overall approach appears incremental rather than conceptually original.

2. The paper’s treatment of the trajectory generation task as an offline RL problem is not rigorously justified. The reward definition, policy learning setup, and connections between the RL objectives and the Transformer pretraining remain vague, raising doubts about whether the RL framing is actually necessary or beneficial.

3. Although quantitative results are reported, improvements over baselines are modest and may fall within normal variance. The paper lacks statistical significance testing, ablation clarity, and detailed comparisons that would substantiate the claim that the reward-model fine-tuning substantially improves reliability and diversity.

4. The interpretability section mainly presents visualizations (e.g., t-SNE and attention maps) without offering meaningful or rigorous analysis. The discussion is largely descriptive and does not provide deeper insight into model behavior, decision rationale, or learned representations.

5. The paper overemphasizes privacy as motivation but does not demonstrate or quantify privacy preservation in any way. There is no clear evaluation of whether the generated trajectories actually protect user privacy or can be used safely for urban applications, undermining the practical relevance and impact of the work.

**Questions:**

1. The paper frames trajectory generation as an offline RL task, but it is unclear why this is necessary. Please clarify how the RL setup (as opposed to pure supervised Transformer training) contributes to performance gains, ideally with ablation or theoretical justification.

2. The IRL-based reward model is central to the framework, yet its construction and validation are underexplained. How is it trained, what signals represent “individual preferences,” and how do you verify that the learned rewards correspond to meaningful mobility behaviors?

3. Report the statistical significance or variance of the results. Are the gains over baselines (especially TrajGPT-DPO) robust across random seeds or datasets? Without such evidence, it is hard to assess whether the improvements are meaningful.

4. The contribution of the reward model–based fine-tuning (RMFT) remains ambiguous. Please provide deeper ablations to isolate the effects of the reward model and discuss sensitivity to key hyperparameters such as the KL penalty

5. Since privacy preservation is used as a major motivation, empirical evidence is needed. Can you evaluate whether the generated trajectories prevent user re-identification or information leakage, or quantify privacy guarantees using standard metrics?

---

### Note · Authors · 2025-11-12

I have read and agree with the venue's withdrawal policy on behalf of myself and my co-authors.